# Rifaximin Improves Liver Functional Reserve by Regulating Systemic Inflammation

**DOI:** 10.3390/jcm12062210

**Published:** 2023-03-13

**Authors:** Kensuke Kitsugi, Kazuhito Kawata, Hidenao Noritake, Takeshi Chida, Kazuyoshi Ohta, Jun Ito, Shingo Takatori, Maho Yamashita, Tomohiko Hanaoka, Masahiro Umemura, Moe Matsumoto, Takafumi Suda

**Affiliations:** Department of Internal Medicine II, Hamamatsu University School of Medicine, Hamamatsu 431-3192, Japan

**Keywords:** albumin, C-reactive protein, Child–Pugh–Turcotte score, cirrhosis

## Abstract

Rifaximin, a non-absorbable antibiotic, has been demonstrated to be effective against hepatic encephalopathy (HE); however, its efficacy on liver functional reserve remains unknown. Here, we evaluated the efficacy of rifaximin on the liver functional reserve and serological inflammation-based markers in patients with cirrhosis. A retrospective study was conducted on patients who received rifaximin for more than three months at our hospital between November 2016 and October 2021. The recurrence and grade of HE, serological ammonia levels, Child–Pugh score (CPS), and serological inflammation-based markers such as the neutrophil–lymphocyte ratio (NLR), lymphocyte–monocyte ratio (LMR), platelet–lymphocyte ratio (PLR), C-reactive protein (CRP), and CRP to albumin ratio (CAR) were evaluated. The correlations between serological inflammation-based markers and liver functional reserve were evaluated. HE grades, serum ammonia levels, and inflammation-based markers significantly improved at three months compared with those at baseline. Patients with improved albumin levels showed significantly higher CRP improvement rates at both 3 and 12 months. Patients with an improvement in CAR at 3 months demonstrated a significant improvement in CPS at 12 months. Rifaximin improved the liver functional reserve in patients with cirrhosis. Improvements in inflammation-based markers, particularly CRP and albumin, may be involved in this process.

## 1. Introduction

Cirrhosis is regarded as the terminal state of chronic liver disease. According to the Global Burden of Disease Project, cirrhosis causes approximately one million deaths annually, making it the 11th leading cause of death globally [1]. The Child–Pugh score (CPS) is often used to evaluate liver functional reserve in patients with cirrhosis [2]. The exacerbation of CPS is correlated with poor prognosis, and the one-year survival rate of patients with a Child–Pugh class of C (CPS ≥ 10) is approximately 45% [3]. In addition, the liver functional reserve also affects the selection of treatment for hepatocellular carcinoma (HCC), and patients with poor liver functional reserve are often forced to abandon HCC treatment [2]. Some cirrhotic patients with hepatitis C viral infection have been reported to have improved liver functional reserves after antiviral therapy [4], but this is limited to a few, and the improvements are insufficient. Therefore, the development of therapeutics focused on improving liver functional reserve is an urgent issue.

Rifaximin is classified as a rifamycin antibiotic like rifampicin and rifabutin and has antimicrobial activity against Gram-positive, Gram-negative, aerobic, and anaerobic bacterial flora via the inhibition of RNA synthesis by binding to the β-subunit of bacterial DNA-dependent RNA polymerase [5,6]. In addition to its direct bactericidal effect, it decreases bacterial virulence factors and morphology [5]. Moreover, it is a safe drug with little absorption from the intestinal tract and few systemic side effects [6].

Rifaximin has been approved for the treatment of irritable bowel syndrome, small intestine bacterial overgrowth, travelers’ diarrhea, and hepatic encephalopathy (HE) [5]. Among these diseases, HE is a typical complication that deteriorates prognosis in patients with cirrhosis [7]. Furthermore, rifaximin has been demonstrated to prevent other complications of cirrhosis, such as esophagogastric variceal bleeding, spontaneous bacterial peritonitis, and hepatorenal syndrome [8,9].

Systemic inflammation plays a crucial role in cirrhosis progression. Patients with advanced cirrhosis have an elevated systemic inflammatory status [10,11]. Increased inflammatory cytokines in patients with cirrhosis exacerbate prognosis by causing protein/energy malnutrition and contributing to liver carcinogenesis [12,13]. Furthermore, several reports have suggested that endotoxemia is associated with the incidence and severity of HE [14,15]. Therefore, patients with HE are expected to demonstrate an elevated systemic inflammatory response. In addition, increasing levels of inflammatory cytokines such as interleukin (IL)-6 and tumor necrosis factor-alpha (TNF-α) trigger the inflammatory immune response and predispose patients with cirrhosis to infectious complications such as sepsis [16]. Thus, systemic inflammation and immune function are inextricably linked, and serological inflammation-based markers that reflect them have been developed; furthermore, their association with prognosis in various diseases has been reported [17,18]. In addition, several inflammation-based markers, such as C-reactive protein (CRP) and neutrophil-to-lymphocyte ratio (NLR), can predict outcomes in patients with cirrhosis [19,20]. Therefore, targeting the patient’s systemic inflammatory status seems to be beneficial.

Previous studies have shown that rifaximin reduces serum levels of endotoxins and inflammatory cytokines such as IL-6 and TNF-α [21]. Furthermore, rifaximin improves the prognosis of patients with alcoholic cirrhosis, where inflammatory cytokines are involved in disease progression [9]. However, the efficacy of rifaximin treatment on liver functional reserve has not been elucidated, and the relationship between systemic inflammatory status and the prognosis of cirrhosis in rifaximin treatment remains unclear. Here, we evaluated the efficacy of rifaximin on serological inflammation-based markers and liver functional reserves in patients with cirrhosis.

## 2. Materials and Methods

### 2.1. Study Population

This retrospective observational study aimed to investigate the efficacy of rifaximin administration on serological inflammation-based markers and liver functional reserve in patients with cirrhosis. Between November 2016 and October 2021, 66 patients with cirrhosis aged ≥ 18 years who began receiving 1200 mg of rifaximin per day (400 mg three times per day) for HE symptoms at Hamamatsu University Hospital were enrolled. Of the enrolled patients, those who discontinued the administration of rifaximin within three months and received other drugs for HE, such as branched-chain amino acids (BCAAs), synthetic disaccharide, zinc preparation, and levocarnitine, within the observation period were excluded. The remaining 44 patients who were evaluable 3 months after rifaximin administration were included in Examination 1, and 29 patients who were evaluable 12 months after rifaximin administration were included in Examination 2 (Figure 1). This study was approved by the Ethics Committee of Hamamatsu University Hospital. Each patient was offered the choice to decline participation in this study via an opt-out option.

### 2.2. Evaluations

Clinical history data for enrolled patients were collected. Hematological and biochemical parameters, including white blood cell count, total neutrophil count, total lymphocyte count, total monocyte count, platelet count, total bilirubin, albumin, CRP, prothrombin time, and ammonia levels, were measured using standard laboratory methods at baseline and 3 and 12 months after the start of rifaximin administration. We evaluated the CPS as a functional reserve of the liver. NLR, lymphocyte–monocyte ratio (LMR), platelet–lymphocyte ratio (PLR), CRP, and CRP to albumin ratio (CAR) were evaluated as serological inflammation-based markers at the time of assessment. These markers were calculated as follows: NLR = total neutrophil count/total lymphocyte count, LMR = total lymphocyte count/total monocyte count, PLR = platelet count/total lymphocyte count, and CAR = serum CRP level/serum albumin level. The classification of HE was based on the West Haven criteria [22]. HE recurrence was defined as grade 1 or higher, and overt HE was defined as grade 2 or higher.

In Examination 1, we investigated the impact of rifaximin on HE, serum ammonia levels, inflammation-based markers, and liver functional reserve three months after initiating rifaximin administration. In Examination 2, we investigated the impact of rifaximin on liver functional reserve and the correlation between inflammation-based markers and liver functional reserve 12 months after initiating rifaximin administration.

### 2.3. Statistical Analyses

Patient characteristics are presented as numbers for categorical data and medians and total ranges for continuous variables. Differences between paired groups were analyzed using the Wilcoxon signed-rank test. We used the Mann–Whitney U test to compare independent samples. Categorical variables of the independent samples were compared using Fisher’s exact test. Cramer’s coefficient of association was calculated to evaluate the strength of the association between the two categorical variables. All analyses were performed using EZR [23], which is used in R. More precisely, it is a modified version of the R commander designed to add statistical functions frequently used in biostatistics. A *p*-value of <0.05 was considered statistically significant.

This study followed the Strengthening the Reporting of Observational Studies in Epidemiology (STROBE) reporting guidelines [24].

## 3. Results

### 3.1. Patient Characteristics

Baseline patient characteristics are shown in Table 1. The median age of the patients was 69 years (range of 34–84 years). The study included 23 males (52%) and 21 females (48%). HE grades at baseline were as follows: grade 1 in 27 cases (62%); grade 2 in 15 cases (34%); grade 3 in 1 case (2%); and grade 4 in 1 case (2%). Approximately 80% of the cases were due to causes other than hepatitis C virus infection, and alcohol abuse was the most common cause of cirrhosis. Regarding liver function, 80% of cases were classified as Child–Pugh class B or C. The median value of CPS was 8 [6,7,8,9,10,11,12,13,14,15]. At baseline, 12 patients (27%) had HCC, 26 (59%) had esophagogastric varices, and 18 (41%) had ascites, 3 of whom had a history of spontaneous bacterial peritonitis (SBP). Within three months, four cases underwent HCC treatment, such as radiofrequency ablation and transcatheter arterial chemoembolization. None of the patients had esophageal varices rupture or SBP expression. Thirty-seven cases (84%) received concomitant medications at the start of rifaximin administration. The median serum ammonia level was 100 μg/dL (24–285 μg/dL). Median values of NLR, LMR, PLR, CRP, and CAR were as follows: NLR, 2.73 (1.00–23.00); LMR, 2.75 (0.89–10.00); PLR, 92.83 (26.09–336.84); CRP, 0.41 mg/dL (0.05–4.55 mg/dL); and CAR, 0.15 (0.014–2.68).

### 3.2. Impact of Rifaximin on Hepatic Encephalopathy and Serum Ammonia Levels

HE recurrence was observed in eight patients (20%) within three months. At three months, HE grades improved in 37 cases (84%), remained unchanged in 6 cases (14%), and were exacerbated in 1 case (2%). The proportion of patients with overt HE significantly decreased from 38% at baseline to 7% at three months. A significant difference in the distribution of HE grades was observed between baseline and three months (*p* < 0.01) (Figure 2A). The median level of serum ammonia was 75.5 μg/dL at baseline, and it significantly decreased to 52.3 μg/dL at three months (*p* < 0.01) (Figure 2B).

### 3.3. Impact of Rifaximin on Serological Inflammation-Based Markers

We investigated the effect of rifaximin on inflammation-based markers. At three months of rifaximin administration, significant improvements in inflammation-based markers, namely CRP (*p* < 0.01) and CAR (*p* < 0.01), were observed. (Figure 3). These results indicate that rifaximin administration improved CRP and CAR at three months in patients with cirrhosis.

### 3.4. Impact of Rifaximin on Liver Functional Reserve at Three Months

Next, we investigated the effect of rifaximin on the liver functional reserve in patients with cirrhosis. At three months, CPS improved in 25 patients (57%), remained unchanged in 14 patients (32%), and was exacerbated in 5 cases (11%). A significant improvement in the median CPS value was observed from 8 (6–15) at baseline to 7 (5–13) at three months (*p* < 0.01) (Figure 4A). The Child–Pugh classes at baseline and at three months are shown in Figure 4B. A total of 15 cases (34%) showed an improvement in the Child–Pugh class, including 8 cases in which there was an improvement from class B to class A and 7 cases from class C to class B. A total of four cases (9.1%), including three cases from class A to class B and one case from class B to class C, showed exacerbation. To examine which parameters contributed to the improvement in CPS, the changes in five parameters of CPS (prothrombin time (PT), total bilirubin, ascites, HE, and albumin) were examined. Figure 4C depicts the distribution of changes in each parameter in the improved CPS group (CPS improvement group, 25 cases) and the unchanged or exacerbated CPS group (CPS non-improvement group, 19 cases). As shown in Figure 2, rifaximin administration had excellent efficacy on HE in cirrhotic patients, but there was no significant difference in HE scores between the two groups. Similarly, no significant differences in PT, total bilirubin, and ascites scores were observed. There was, however, a significant difference in the rate of albumin score improvement between the CPS improvement and CPS non-improvement groups (*p* < 0.01). In Cramer’s coefficient of association, changes in albumin scores showed the most significant correlation with changes in CPS compared to other parameters (albumin, v = 0.41; PT, v = 0.25; total bilirubin, v = 0.05; ascites, v = 0.22; and HE, v = 0.12). When the CRP improvement rate in the albumin improvement group was compared with that in the albumin non-improvement group, the albumin improvement group showed a significantly higher CRP improvement rate (*p* < 0.05) (Figure 4D).

These results suggest that rifaximin improves CPS through the improvement in albumin and that the improvement in CRP may be involved in this process. 

### 3.5. Investigation of the Effect of Rifaximin on Liver Functional Reserve at 12 Months and Its Relation to Inflammation-Based Markers

We next investigated the effect of rifaximin on liver functional reserve at 12 months (Examination 2). The background characteristics of the 29 patients are presented in Table 2. At 12 months, CPS improved in 16 cases (55%), remained unchanged in 10 cases (34%), and was exacerbated in 3 cases (10%) compared to that at baseline. The median CPS value showed significant improvement from 8 (6–14) at baseline to 7 (5–11) at 12 months (*p* < 0.01) (Figure 5A). As shown in Figure 5B, a total of nine cases (31%) showed an improvement in Child–Pugh class at 12 months, including four cases showing an improvement from class B to class A and five cases from class C to class B. However, one case (3.4%) was exacerbated from class A to class B. Similar to the parameters that contributed to the improvement in CPS in Examination 1, the ratio of the improvement in albumin score was significantly higher in the CPS improvement group (16 cases) than in the CPS non-improvement group (13 cases) (*p* < 0.05) (Figure 5C). In Cramer’s coefficient of association, changes in albumin showed the most significant correlation with changes in CPS compared to other parameters (albumin, v = 0.46; PT, v = 0.17; total bilirubin, v = 0.17; ascites, v = 0.23; and HE, v = 0.04). Moreover, we compared the CRP improvement rate in the albumin improvement and non-improvement groups at 12 months. Similar to Examination 1, the CRP improvement rate was significantly higher in the albumin improvement group (*p* < 0.05) (Figure 5D).

Further investigation focused on the association of changes in CPS at 12 months after rifaximin administration with serological inflammation-based markers at 3 months. Interestingly, a significant increase in the improvement rate of CPS at 12 months was noted in the CAR improvement group at 3 months (*p* < 0.05) (Figure 5E). In contrast, there was no significant difference in the changes in CRP at three months between the two groups. These results indicate that CAR 3 months after rifaximin administration may predict CPS improvement at 12 months.

Taken together, our results suggest that rifaximin may improve serum CRP and albumin levels, which are crucial indicators of immunity and inflammation, and contribute to improving the liver functional reserve.

## 4. Discussion

This study evaluated the efficacy of rifaximin on the liver functional reserve and systemic inflammation in patients with cirrhosis. Rifaximin administration for more than three months improved CPS by improving serum albumin levels, suggesting that the improvement in systemic inflammation, especially CRP, was involved in this process. Meanwhile, CPS worsened at three months after rifaximin administration in 5 of 44 cases. CRP levels increased at three months in all four cases except one case where the patient died of progressing liver failure at four months of rifaximin administration. Thus, these results also suggest an association between CRP and liver functional reserve. Moreover, two cases in which CPS worsened at 3 months showed improvements in CPS at 12 months, whereas the other two cases could not be evaluated at 12 months. The long-term administration of rifaximin may contribute to the improvement in the liver functional reserve regardless of its short-term effects. However, there are a few reports on the long-term administration of rifaximin, and most of the previous reports are on administration for less than three weeks. In these short-term studies, the liver functional reserve was either not investigated or showed no improvement [25,26,27]. Kalambokis et al. reported that eight weeks of rifaximin administration significantly improved CPS and model for end-stage liver disease scores in cirrhotic patients awaiting liver transplantation [28]. However, this study was limited to patients with alcohol-related cirrhosis, and the number of patients was small [28]. Reports on the long-term administration of rifaximin for more than six months have verified its effect on HE; however, its effect on liver functional reserve remains unknown [29,30]. Therefore, the current study is significant, showing that the administration of rifaximin improved liver functional reserve regardless of etiology, with a larger sample size and longer duration than previous reports.

Several reports have shown an improvement in liver functional reserve with medical treatment in patients with cirrhosis. BCAAs, which are also often administered to cirrhotic patients, have been reported to improve liver functional reserve and contribute to an improved prognosis [31]. Symbiotics also improve CPS by increasing serum bilirubin, serum albumin, and PT levels [32]. Paromomycin improved CPS by improving serum albumin levels [33], and neomycin improved CPS by improving ascites and HE [34]. These reports are similar to our study in that both drugs are non-absorbable antibiotics, similar to rifaximin, and were administered for a relatively long-term course of three–six months [33,34]. In our study, the improvement in serum albumin levels contributed to the improvement in CPS by rifaximin, which is similar to the reports on symbiotics and paromomycin [32,33]. These reports suggested that the alleviation of endotoxemia caused by intestinal bacteria contributed to the improvement in CPS [32,33].

Increased permeability of the intestinal mucosa and disruption of barrier function are observed in patients with cirrhosis, which induce bacterial translocation, enhance the migration of intestinal bacteria, and lead to elevated blood endotoxin levels [35]. Furthermore, endotoxemia causes deterioration of liver functional reserve in cirrhotic patients [36] and is associated with complications, such as esophageal varices and HE [37,38]. There are several reports suggesting that rifaximin improves endotoxemia, and gut microbial changes and improved intestinal permeability are speculated to be the underlying mechanisms [39,40]. The gut microbiota is composed of a large number of different microorganisms in a complex manner and contains not only bacteria but also many viruses, fungi, and mycetes. Additionally, it has been suggested that certain viruses may cause dysbiosis and endotoxemia, exacerbating the disease state [41,42]. However, there is also a report that rifaximin is effective against dysbiosis associated with viral infection and optimizes intestinal permeability [42]. Considering that it is also effective for diseases in which the microbiome is deeply involved, such as irritable bowel syndrome and small intestine bacterial overgrowth [5], there is strong evidence that rifaximin modulates gut microbiota metabolism.

Moreover, albumin can inactivate endotoxins, and there is a correlation between decreased serum albumin levels and increased blood endotoxin levels in cirrhotic patients [43]. The levels of inflammatory cytokines in the blood are also elevated, similar to endotoxin levels in cirrhotic patients [10,11]. Inflammatory cytokines promote liver fibrosis [44], induce protein/energy malnutrition [12], and lead to liver carcinogenesis [13], thereby leading to the deterioration of liver functional reserve and the exacerbation of prognosis. Therefore, inflammatory cytokines are being investigated as crucial therapeutic targets in patients with cirrhosis [45]. In vitro, rifaximin activates the human pregnane X receptor, leading to the inhibition of nuclear factor-kappa B-dependent inflammatory pathways, including TNF-α, IL-6, IL-8, and IL-10 secretion in vitro [46]. Furthermore, the administration of rifaximin for 4–12 weeks has been shown to decrease the blood levels of IL-6, IL-10, and TNF-α in patients with nonalcoholic fatty liver disease and cirrhosis with HE [47,48,49].

Thus, endotoxins and inflammatory cytokines are strongly associated with the liver functional reserve in cirrhotic patients, and the reduction in endotoxins and cytokines may lead to an improvement in liver functional reserve. However, blood endotoxin and inflammatory cytokine levels are not widely measured as indicators. Therefore, we investigated the effects of rifaximin using inflammation-based markers calculated from easily measurable test items, such as CRP, white blood cells, platelets, and albumin. The importance of NLR and LMR in cirrhosis has been previously reported [20,50]. In our study, the three-month administration of rifaximin did not significantly improve the NLR and LMR but did significantly improve CRP levels and CAR (Figure 3). CRP levels are higher in patients with cirrhosis than in healthy subjects and are involved in decompensation and mortality [51]. An increase in IL-6 and TNF-α levels in patients with cirrhosis [10,11] is involved in CRP production [52], suggesting that changes in CRP are linked to changes in these inflammatory cytokines. Serum albumin level is a component of CPS and is a crucial parameter in patients with cirrhosis [2]. Hypoalbuminemia exacerbates the prognosis of patients with cirrhosis [52]. Therefore, CAR, a combined marker of CRP and albumin, is suggested to be a crucial indicator in patients with cirrhosis. CAR has been reported to be useful in predicting the prognosis of HCC treatment [53,54], whereas few reports exist on the usefulness of CAR in patients with cirrhosis [55]. Considering the strong correlation between albumin and inflammatory cytokines, CAR is a useful indicator for patients with cirrhosis. Regarding the effect of rifaximin on CRP and CAR, rifaximin improves CRP levels in small intestinal bacterial overgrowth [56], whereas no studies have examined CRP or CAR in cirrhosis. However, considering that rifaximin improves inflammatory cytokines such as IL-6 and TNF-α [46,47,48,49], there is the possibility that CRP, produced by these cytokines, is also improved. Indeed, we demonstrated that rifaximin improved CPS and that CRP and albumin might play crucial roles in this process.

This study has some limitations. First, this was a single-center retrospective study, and the number of patients was insufficient for multivariate analysis. Moreover, the possibility of selection bias cannot be denied because only 29 patients were evaluable at 12 months after initiating rifaximin administration. Therefore, a prospective randomized controlled trial with larger samples from multiple centers is required in the future. Second, we did not examine the efficacy of rifaximin with regard to the etiology of cirrhosis because the etiology of the target patients varied. In alcoholic cirrhosis, inflammatory cytokines and endotoxemia are closely associated with disease progression [22,45], suggesting that rifaximin may be more effective. Third, factors other than rifaximin may improve liver functional reserve. In this study, to examine the effects of rifaximin, we excluded cases in which other drugs for HE treatment, including BCAAs, were newly started within the observation period after rifaximin administration. Many patients were already taking BCAAs at the time of rifaximin administration, and the duration of BCAA administration varied. BCAAs also improve liver functional reserve [31], and the effect of long-term administration of BCAAs on improving liver functional reserve cannot be ruled out. Fourth, it is hypothetical that rifaximin improves liver functional reserve by improving systemic inflammation in patients with cirrhosis because inflammatory cytokines such as IL-6 and TNF-αand endotoxins were not evaluated in this study. It is necessary to evaluate these inflammatory mediators in the future.

## 5. Conclusions

Rifaximin is a drug with many potential benefits beyond HE. This study demonstrated that rifaximin may improve liver functional reserve in patients with cirrhosis by improving systemic inflammation. CRP and albumin are closely related in patients with cirrhosis. Therefore, CRP and related inflammatory cytokines may be key targets for improving liver functional reserve in patients with cirrhosis. The improvement in the liver functional reserve by targeting systemic inflammation is unique, and we believe it is a therapeutic target that should be examined in the future.

## Figures and Tables

**Figure 1 jcm-12-02210-f001:**
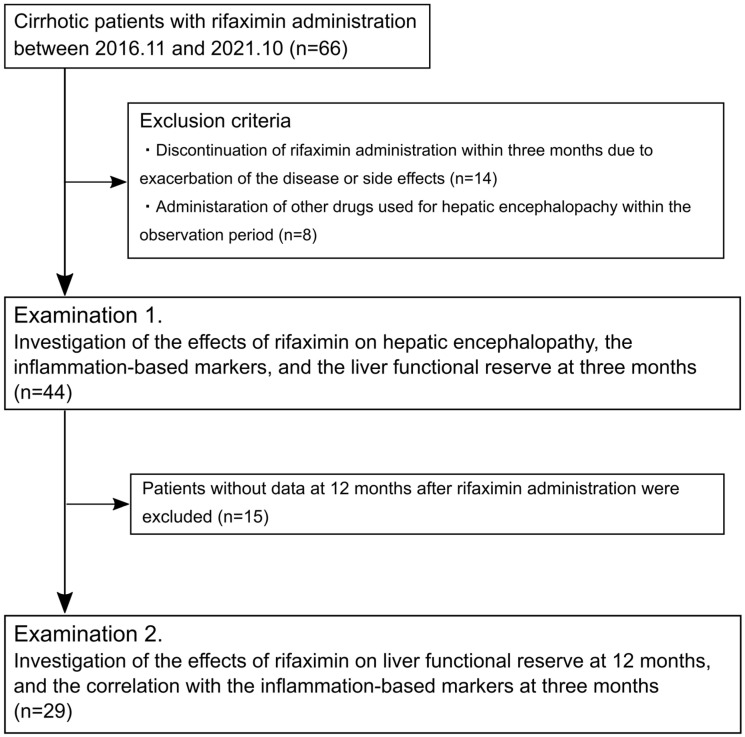
Flow diagram of the study.

**Figure 2 jcm-12-02210-f002:**
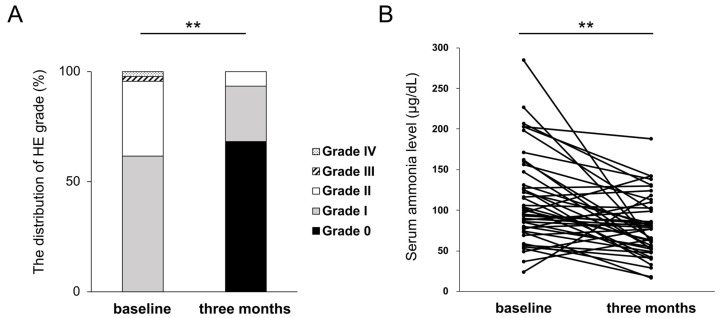
Changes in HE grades (**A**) and serum ammonia level (**B**) at three months of rifaximin administration. ** *p* < 0.01.

**Figure 3 jcm-12-02210-f003:**
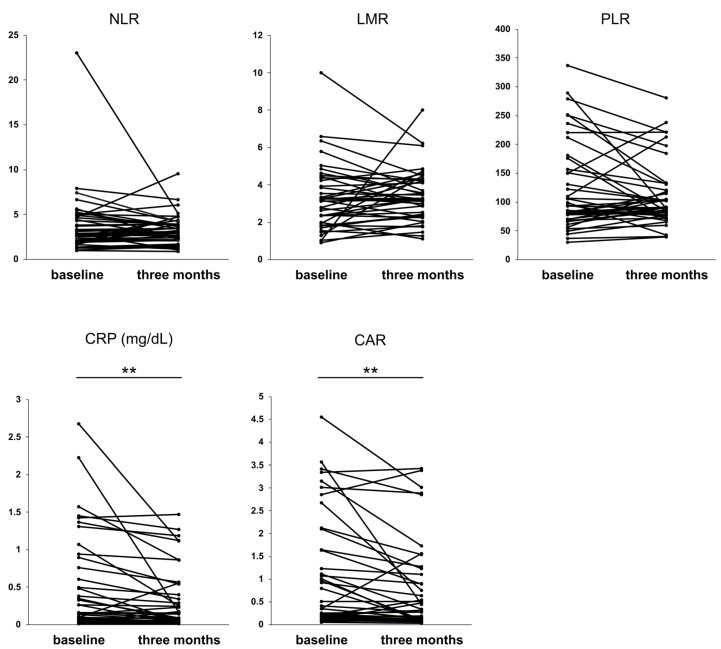
Values of the inflammation-based markers at three months of rifaximin administration. ** *p* < 0.01. NLR, neutrophil–lymphocyte ratio; LMR, lymphocyte–monocyte ratio; PLR, platelet–lymphocyte ratio; CRP, C-reactive protein; CAR, CRP to albumin protein.

**Figure 4 jcm-12-02210-f004:**
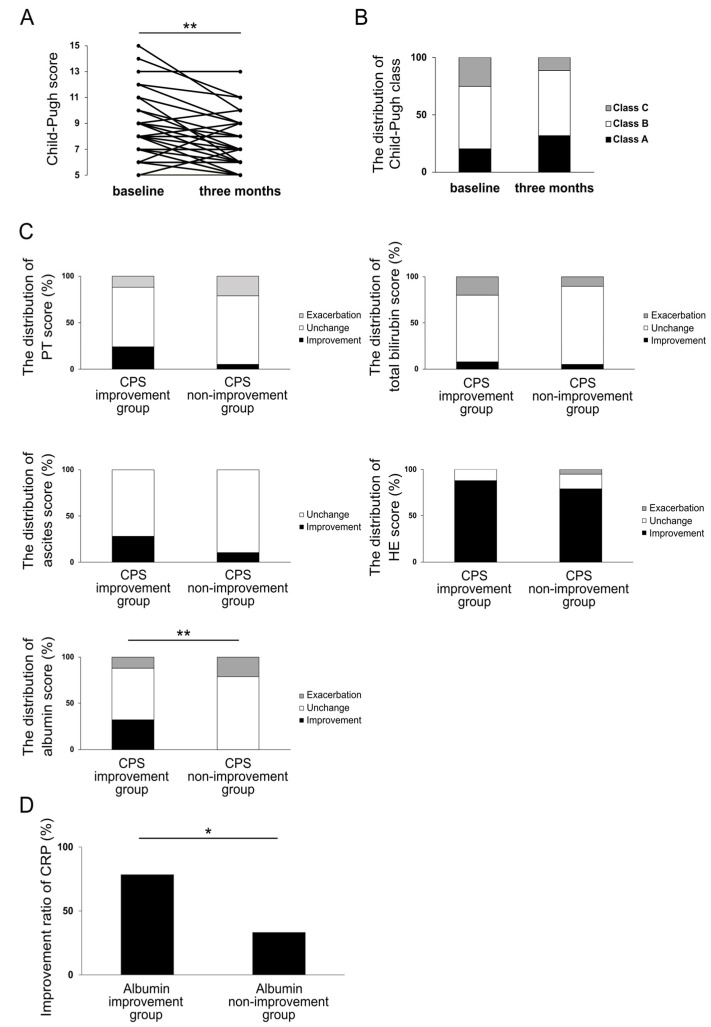
CPS (**A**) and the distribution of Child–Pugh class (**B**) at three months of rifaximin administration. The distribution of changes in CPS parameters in CPS improvement group and CPS non-improvement group at three months of rifaximin administration (**C**). The distribution of changes in CRP in albumin improvement group and albumin non-improvement group at three months of rifaximin administration (**D**). * *p* < 0.05, ** *p* < 0.01. CPS, Child–Pugh score; CRP, C-reactive protein.

**Figure 5 jcm-12-02210-f005:**
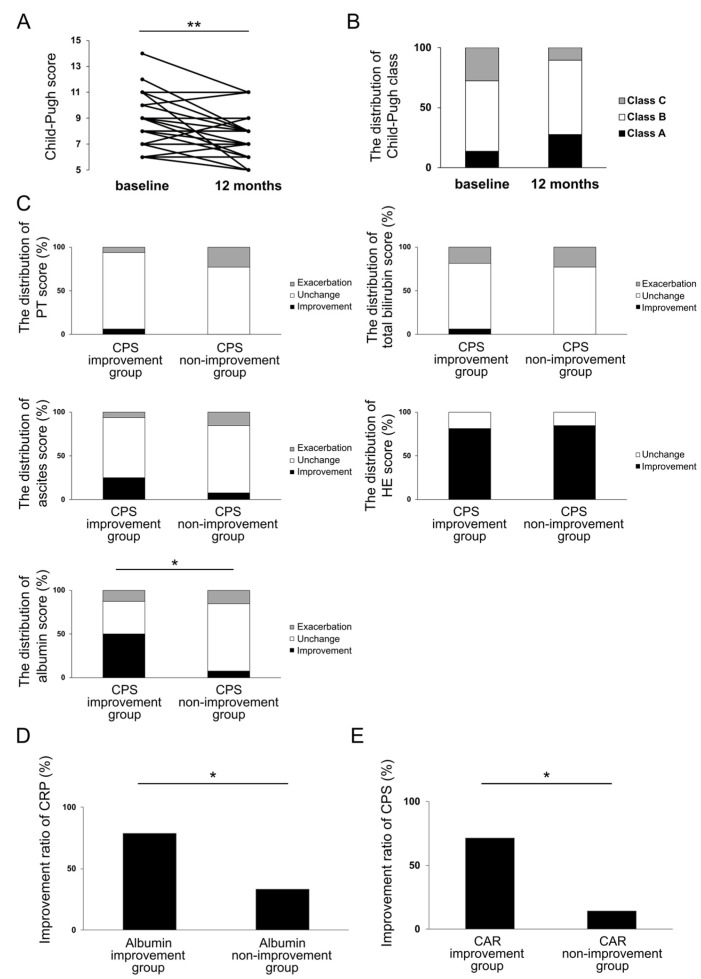
CPS (**A**) and the distribution of Child–Pugh class (**B**) at 12 months of rifaximin administration. The distribution of changes in CPS parameters in CPS improvement group and CPS non-improvement group at 12 months of rifaximin administration (**C**). Improvement ratio of CRP levels in albumin improvement group and albumin non-improvement group at 12 months of rifaximin administration (**D**). Improvement ratio of CPS at 12 months of rifaximin administration in CAR improvement group and CAR non-improvement group at 3 months (**E**). * *p* < 0.05, ** *p* < 0.01. CAR, CRP to albumin ratio; CPS, Child–Pugh score; CRP, C-reactive protein.

**Table 1 jcm-12-02210-t001:** Clinical characteristics of the patient at baseline (Examination 1).

Variable	Results
Age in years	69 (34–84)
Male gender, n (%)	23 (52)
HE grade 1:2:3:4, n (%)	26:12:4:2 (59:27:9:5)
Etiology of liver disease	
Alcohol, n (%)	14 (32)
NASH, n (%)	12 (27)
HCV, n (%)	9 (21)
Autoimmune, n (%)	8 (18)
Cryptogenic, n (%)	1 (2)
Liver functional reserve	
Child–Pugh score	8 (6–15)
Child–Pugh class A:B:C, n (%)	9:24:11 (20:55:25)
Complication in illnesses	
HCC, n (%)	12 (27)
Esophagogastric varices, n (%)	26 (59)
Ascites, n (%)	18 (41)
Concomitant drug *	
BCAA, n (%)	29 (66)
Lactulose or Lactitol, n (%)	15 (34)
Zinc preparation, n (%)	6 (14)
Levocarnitine, n (%)	5 (11)
Inflammation-based markers	
NLR	2.73 (1.00–23.00)
LMR	2.75 (0.89–10.00)
PLR	92.83 (26.09–336.84)
CRP in mg/dL	0.41 (0.05–4.55)
CAR	0.15 (0.014–2.68)

Data are given in median and total range. * There is duplication. Abbreviations: HE, hepatic encephalopathy; NASH, nonalcoholic steatohepatitis; HCV, hepatitis C virus; HCC, hepatocellular carcinoma; BCAA, branched-chain amino acids; NLR, neutrophil–lymphocyte ratio; LMR, lymphocyte–monocyte ratio; PLR, platelet–lymphocyte ratio; CRP, C-reactive protein; CAR, CRP to albumin ratio.

**Table 2 jcm-12-02210-t002:** Clinical characteristics of the patient at baseline (Examination 2).

Variable	Results
Age in years	69 (41–84)
Male gender, n (%)	17 (59)
HE grade 1:2:3:4, n (%)	15:12:1:1 (52:42:3:3)
Etiology of liver disease	
Alcohol, n (%)	10 (35)
NASH, n (%)	8 (28)
HCV, n (%)	5 (17)
Autoimmune, n (%)	5 (17)
Cryptogenic, n (%)	1 (3)
Liver functional reserve	
Child–Pugh score	8 (6–14)
Child–Pugh class A:B:C, n (%)	4:17:8 (14:59:27)
Complication in illnesses	
HCC, n (%)	10 (37)
Esophagogastric varices, n (%)	16 (55)
Ascites, n (%)	10 (37)
Concomitant drug *	
BCAA, n (%)	23 (79)
Lactulose or Lactitol, n (%)	9 (31)
Zinc preparation, n (%)	5 (17)
Levocarnitine, n (%)	6 (21)
Inflammation-based markers	
NLR	2.52 (1.00–7.4)
LMR	3.14 (1.03–10.00)
PLR	81.93 (26.09–336.84)
CRP in mg/dL	0.38 (0.07–4.55)
CAR	0.13 (0.023–2.68)

Data are given in median and total range. * There is duplication. Abbreviations: HE, hepatic encephalopathy; NASH, nonalcoholic steatohepatitis; HCV, hepatitis C virus; HCC, hepatocellular carcinoma; BCAA, branched-chain amino acids; NLR, neutrophil–lymphocyte ratio; LMR, lymphocyte–monocyte ratio; PLR, platelet–lymphocyte ratio; CRP, C-reactive protein; CAR, CRP to albumin ratio.

## Data Availability

The datasets generated and/or analyzed in this study are available from the corresponding author upon reasonable request.

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
