# Peer review of "Rifaximin Improves Liver Functional Reserve by Regulating Systemic Inflammation"

_jcm, 2023, doi:10.3390/jcm12062210_

Round 1

Reviewer 1 Report

Review Report Form

Comments and Suggestions for Authors

The authors Kenbsuke Kitsugi et al., entitled “ Rifaximin improves liver functional reserve by regulating systemic inflammation” show a retrospective study of a case series of patients with liver cirrhosis who took rifaximin over a protracted period of time. The authors illustrate d very well the likely mechanism of action, and the pathways of interference, and highlight important data. Themanuscript is admirable. There are just a few things the authors can do to make it even more robust, and that I recommend. The images all need to be improved because they are of low resolution. Below are my comments.

 Major revisions:

1.       Line 71: please add supplementary material with  the strobe document, https://www.strobe-statement.org/

2.       Line 85: flow diagram, please insert an image of high resolution or create the flow chart on a word file directly

3.       Line 101: it would be important to know the dosages of rifaximin used and whether the authors considered that more aggressive therapies could lead to the same results

4.       Line 149 and 158, 189, and 232 the quality of the figures need to be higher, please insert images at 300 dpi RGB.

5.       Line 315: I really appreciate the authors' honesty in describing their limitations. It is true that they cannot have a multivariate analysis, but I suggest they try to do it anyway just to check for possible confounders. Please if only is possible insert this data in supplementary materials and explain well that it is not significant but serves as a basis for future studies. If they can't, maybe they will do it in another paperm, next time.

6.       Please extend the conclusion.

Minor revision

1.       Line 40: Make the introduction more robust and increase notions of the local mechanisms of action of rifaximin, what type of bacteria are the  target and with which mechanism.

2.       Line 165 and 200: what do the authors think are the causes of why some cases have worsened? It would be good to highlight at least some hypotheses.

3.       Line 267:  The authors  considered  that rifaximin being a primarily local-acting antibiotic, also acts on bacterial endotoxemia of the microbiome. One of their comments as a hypothesis would be indicated.  In this hypothesis, they may also consider viral pathologies or dysbiosis such as.: Brogna, C.; Cristoni, S.; Brogna, B.; Bisaccia, D.R.; Marino, G.; Viduto, V.; Montano, L.; Piscopo, M. Toxin-like Peptides from the Bacterial Cultures Derived from Gut Microbiome Infected by SARS-CoV-2—New Data for a Possible Role in the Long COVID Pattern. Biomedicines 202311, 87. https://doi.org/10.3390/biomedicines110100

Advice for the authors:

Rifaximin is one of the few antibiotics that act locally. These observations of yours are very important. You have evaluated its action and correlations on blood markers. Have you by any chance, considered doing microbiome analysis with 16s sequencing or proteomics with mass spectrometry? If yes, it would be really great to look at them; if not, I invite you to do it soon for an upcoming paper. The toxicology of the microbiome, not known for genetic changes or hidden viral pathologies, represents the target of the future. I advise the authors to continue their work, not to exclude anything, and to have an open mind, as they are showing every signal or sign they can find in the human microbiome. For future studies, I suggest also evaluating the level of pseudocholinesterases, acetyl, and butyryl cholinesterases, which are linked to toxicological events.

Reviewer 2 Report

The manuscript is very interesting. Just like the Authors, I did not find similar studies analyzing long-term rifaximine administration and the impact on the liver condition in terms of albumin synthesis and Child-Pugh Scoring. I did not find any major issues in the paper, only one small, editorial actually: lines 98-99: there is a sentence “The classification was based on the West Haven criteria [21]”. The previous sentence is related to blood sample analysis, and I believe it would be good to rephrase the above sentence to “The classification of HE was based on the West Haven criteria [21]” or similar to point out that the topic has changed to encephalopathy. 

Reviewer 3 Report

The retrospective observational study by  Kensuke Kitsugi et al reports biochemical and clinical data of a group of patients with cirrhosis tested at time 0, after 3 months (44 subjects) and after 12 months (29 subjects). Across an context of the unspecified myriad of conditions and treatments that characterises the clinical story of patients with advanced cirrhosis (the majority where class B/C patients, 32% had ALD and 27% HCC), the patients received rifaximin 1200 mg/day.

Many indexes improved during follow up.

 Major criticism

The design of the study prevents from attributing any of the changes to rifaximin.  

The plots prevent from understanding the change in each patients.

Round 2

Reviewer 1 Report

The authors have improved the manuscript. Below are minor revisions.

Minor revision

1. Line 40-  please, add the pharmaceutical class of Rifaximin and a small list of the other antibiotics in the same class, e.g. rifampicin, etc

2. line 332, please also add "fungi and mycetes", near the word "bacteria" 

3. line 466. please  replace "function" with " metabolism"

Thank you to the authors for answering my questions.
